# Hypochlorous Acid as a Potential Postsurgical Antimicrobial Agent in Periodontitis: A Randomized, Controlled, Non-Inferiority Trial

**DOI:** 10.3390/antibiotics12081311

**Published:** 2023-08-12

**Authors:** Julio Cesar Plata, David Díaz-Báez, Nathaly Andrea Delgadillo, Diana Marcela Castillo, Yormaris Castillo, Claudia Patricia Hurtado, Yineth Neuta, Justo Leonardo Calderón, Gloria Inés Lafaurie

**Affiliations:** 1Master’s Program in Dental Sciences, School of Dentistry, Universidad El Bosque, Bogotá P.O. Box 110121, Colombia; julcepla@hotmail.com; 2School of Dentistry, Universidad Cooperativa de Colombia, Bucaramanga P.O. Box 680001, Colombia; od.claudiah@hotmail.com; 3Unit of Oral Basic Investigation-UIBO, School of Dentistry, Universidad El Bosque, Bogotá P.O. Box 110121, Colombia; dadiazb@unbosque.edu.co (D.D.-B.); ndelgadillos@unbosque.edu.co (N.A.D.); castillodiana@unbosque.edu.co (D.M.C.); castilloyormaris@unbosque.edu.co (Y.C.); yneuta@unbosque.edu.co (Y.N.); justo05@gmail.com (J.L.C.)

**Keywords:** chlorhexidine, antiplaque, bacterial recolonization, hypochlorous acid

## Abstract

Background: Hypochlorous acid (HOCl) is an antimicrobial agent with high affinity to Gram-negative bacteria of the subgingival biofilm. It could have an equivalent or no inferiority effect to chlorhexidine (CHX) to avoid recolonization of these microorganisms after the post-surgical period. Objective: The objective is to compare the reduction of plaque index (PI), gingival index (GI), pocket depth (PD), gain of clinical attachment level (CAL), and bacterial recolonization of periodontopathic microorganisms in subgingival biofilm at 7, 21, and 90 days after Open Flap Debridement (OFD) under two antimicrobial protocols: (A) HOCl 0.05% followed by HOCl 0.025% and (B) CHX 0.2%/CHX 0.12% used per 21 days without regular oral hygiene during the post-surgical period. Material and methods: A no-inferiority randomized controlled trial was carried out. Thirty-two patients were randomly divided to receive each antiplaque protocol after OFD in patients with periodontitis. Clinical indexes and bacterial recolonization were assessed using qPCR for up to 90 days. Data were analyzed using repeated measures ANOVA, mixed effects models adjusted for treatment, time, and the Chi-squared/Fisher test. A no-inferiority analysis was also performed using the Hodges–Lehmann hypothesis test for non-inferiority. Results: HOCl was not inferior to CHX in reducing PI. Both groups showed a comparable reduction of recolonization for *Porphyromonas gingivalis*, *Tannerella forsythia*, and *Eubacterium nodatum*. However, the HOCl protocol was non-inferior to the CHX protocol for *Treponema denticola* and *Aggregatibacter actinomicetemcomitans*. Conclusions: HOCl improved periodontal healing. HOCl showed an impact in reducing the recolonization of periodontopathic bacteria in the postoperative period.

## 1. Introduction

Periodontitis is a multifactorial inflammatory disease associated with biofilm dysbiosis [1]. However, although the disease remains stable after periodontal therapy, its progression occurs in sites related to poor oral hygiene [2].

Chlorhexidine (CHX) rinses minimize biofilm formation and gingival inflammation following periodontal surgery. However, the impact of reducing the periodontal probing depth (PD) is unclear [3]. Furthermore, CHX reduces bacterial recolonization after periodontal surgery, which favors healing and avoids the recurrence of the periodontal lesion [4]. Although CHX is safe, stable, and effective in minimizing periodontal pathogen recolonization and preventing biofilm formation, several side effects, such as dental surface pigmentation, taste modification, scaly lesions on the mucosa, dryness of the tissues, and periodontal healing delay, among others, have limited its clinical use [5].

Neutrophils and macrophages synthesize hypochlorous acid (HOCl) during phagocytosis of antigens as the final product of H_2_O_2_ by the action of the myeloperoxidase and Cl_2_, and this is synthesized and stabilized to use in clinical medicine for skin infections, burn wound healing, and chronic leg ulcers [6,7]. HOCl is effective against many Gram-negative microorganisms recognized as periodontal pathogens using concentrations between 180 and 500 ppm [8,9] or low concentrations combined with stabilized acetic acid to reduce bacterial viability on oral biofilm [10]. Moreover, HOCl is an oxidizing agent with an excellent viricidal effect, including SARS-CoV-2 [11]. HOCl displays low toxicity and anti-inflammatory and proliferative cell effects. It is a promising molecule for post-surgical periodontal use [6].

The primary objective of this study was to compare the clinical and microbiological efficacy of postsurgical protocols with HOCl at 0.05%/0.025% and CHX at 0.2%/0.12% as antimicrobial agents in patients with chronic periodontitis following 7, 21, and 90 days of surgical periodontal therapy. Adverse effects were also evaluated.

## 2. Results

### 2.1. Characteristics of the Study Population

Thirty-two individuals were examined and randomly divided into two antiplaque protocols of sixteen each. Both protocols were strictly conducted for 21 days, following open flap debridement (OFD): (A) HOCl 0.05% (from day 0 to 7) followed by HOCl 0.025% (from day 7 to 21) and (B) CHX 0.2% (from days 0 to 7) followed by CHX 0.12% (days 7 to 21). Both groups were clinically and microbiologically examined at baseline, 7, 21, and 90 days (Figure 1). At baseline, gender, age, number of teeth, and clinical index were statistically comparable between both groups; thus, the groups were similar since the beginning of the study (Table 1). The percentages of initial probing depths ≥ 5 mm for the CHX and HOCl groups were 13.7% and 13.8%, respectively (*p* > 0.05). Similarly, the percentages of initial probing depths greater than 6 mm for the CHX and HOCl groups were 8.25% and 7.56%, respectively (*p* > 0.05).

### 2.2. Clinical Indexes over Time

Table 2 compares the plaque and gingival indexes (GI) between the two protocols (CHX 0.2/0.12%/HOCl 0.05/0.025%) over time. When the plaque index (PI) was compared, significant differences were found between the baseline (t_0_) and times t_1_ (day 7), t_2_ (day 21), and t_3_ (day 90) (*p* < 0.001) in both groups. The most significant plaque reduction was observed when comparing t_0_ vs. t_2_, for the CHX group, with 10% more dental plaque in the HOCl (22 ± 15 vs. 12 ± 7); however, the no-inferiority hypothesis between groups was verified (*p* < 0.05). In GI, differences were only found between pretreatment and the other times for both groups (*p* < 0.001) but not between groups (Table 2). Periodontal pocket depth (PD), bleeding probing (BoP) reduction, and gain of clinical attachment level (CAL) between postsurgical protocols over time are shown in Table 3. In the two groups, statistical differences were observed when t_0_ was compared with t_3_ (day 90) (*p* < 0.001). The differences between the groups were slight. However, in this study, verifying equivalence between the groups was impossible.

The attributable risk reduction (ARR) is observed in Figure 2. CHX reduced the ARR for plaque index (PI) at 7 and 21 days. At 90 days after OFD, both groups were similar for gingivitis reduction, PI, and bleeding on probing. For gain of CAL > 3 mm and reduction of PD > 6 mm at 90 days, HOCl shows better performance.

Table 4 shows the concentration of the microorganisms evaluated. *P. gingivalis* was the most frequent microorganism in the pretreatment, followed by *T. forsythia*, *T. denticola*, and *E. nodatum*. *A. actinomycetemcomitans* presented the lowest in both groups. The changes in the concentrations of microorganisms were carried out with a linear model of mixed effects of repeated measures and the Hodges–Lehmann Test hypothesis test for non-inferiority between groups. All microorganisms showed a reduction in the colony-forming units by milliliters (UFC/mL) between t_0_ and all times except for *A. actinomycetemcomitans* which only showed reductions at t_1_. When comparing the two protocols, only statistical differences were observed for *P. gingivalis* at t_2_, with a no-inferiority reduction between treatments. The count of the other microorganisms was meager at 90 days.

In general, a tendency to the recolonization of this microorganism was achieved at 90 days in both groups. Table 5 shows a significant reduction in *P. gingivalis* between times t_0_ to t_1_, t_2_, and t_3_ (*p* < 0.001) in both groups. CHX protocol showed a decrease of 43% with recolonization of 50%; only a low level of non-detection was observed in the patients at 90 days (12%). In the HOCl group, the reduction of *P. gingivalis* was 46.7%, recolonization of 46.7%, and only a level of non-detection in 6% of the patients at 90 days. The recolonization was similar when comparing the two groups.

For *T. forsythia* and *T. denticola*, the t_0_ detection frequency was 75% for the CHX group and 81.2% for the HOCl group. The recolonization was more significant in the CHX protocol than HOCl (62% vs. 46.6%) at 90 days. However, the no-inferiority hypothesis can be demonstrated only for *T. denticola* in t_2_ (*p* < 0.02).

*A. actinomycetemcomitans* detection frequency was 50% for the CHX group and 68.7% for HOCl at baseline. The reduction of *A. actinomycetemcomitans* was significant for all times in both groups (*p* < 0.05); the difference was demonstrated to be no-inferiority at t_1_ and t_2_ for recolonization (*p* = 0.10) and a reduction in t_1_ (*p* = 0.016) and t_2_ (*p* = 0.029). The recolonization at 90 days was 18.7%, with a decrease in the microorganism concentration of 37.5% in CHX, and 40% of recolonization, with a 26.6% reduction in HOCl; however, the no-inferiority hypothesis at 90 days cannot be demonstrated. For *E. nodatum*, the detection frequency at time t_0_ was slightly lower than the other anaerobic microorganisms; the CHX group was 56.2%, and the HOCl group was 68.7%. This microorganism showed the highest recolonization at t_3_ (90 days) of treatment in both groups and showed rapid recolonization in the two groups (Table 5).

### 2.3. Adverse Effects

Table 6 resumes the adverse effects at 7 and 21 days of follow-up. HOCl showed more unpleasant sensations than CHX rinses. At t_2_ (day 21), the HOCl rinse led to an unpleasant feeling for 93.7% of the participants, CHX was 56.2%, and a statistical difference was demonstrated between the groups (*p* = 0.048) (superiority hypothesis). CHX rinses presented more irritation sensation than HOCl at 21 days. CHX showed more sensation of dryness at 7 days; however, this was reduced at 21 days without observing differences between the groups. No group patient reported pain during the study; only one patient with CHX reported burning at t_1_ and t_2_ (Table 6).

The burning sensation was more remarkable for the 0.2% CHX rinse than HOCl on day 7. At 21 days, CHX 0.12% and HOCl 0.025% rinses showed the same burning sensation.

The sensation of roughness and gastric alteration was similar for the two protocols. In both rinses, the most reported discomfort was in the lips, especially with 0.05% HOCl in t_1_. At 21 days, no patient reported discomfort in the lips. The CHX 0.2% rinse reported the most significant change in taste sensation on days 7 (*p* = 0.049) and 21 days (*p* = 0.009).

The patients reported a change in color since day 7 in both groups. At 21 days, 68.75% for CHX and 31.25% for HOCl presented color changes sensation (*p* = 0.034). When asking patients, the reported color change for CHX was black or brown spots. In contrast, the perceived color change for HOCl was towards whitening or “whiter teeth”. Only one patient reported dental sensitivity in the HOCl group on day 21 (Table 6).

No growth of opportunistic microorganisms in saliva was observed in any groups during the observation time.

## 3. Discussion

According to the findings of this study, the 0.2%/0.12% CHX antiplaque treatment resulted in significant reductions in PI of roughly 65% at 7 and 21 days post-surgical periods. Comparable results were reported by previous post-surgical studies [12]. HOCl also showed a reduction above 50% over time. CHX is more effective by 10% in reducing PI than HOCl at seven days, although after 21 days, individuals with non-regular hygiene are similar. Nevertheless, the HOCl protocol is not inferior in reducing PI because the limit of non-inferiority in this study was demonstrated. The efficacy of GI, BoP, and PD at 21 days post-surgery was similar between protocols. However, at 90 days, HOCl performed better than CHX in CAL gain > 3 mm. These results could be explained by the significant effect of HOCl on the gram-negative microorganisms and the anti-inflammatory and proliferative cell effects during the healing [7,8,9,10,11].

Bacterial colonization of tooth surfaces is a relevant cause of periodontitis recurrence. Bacterial antimicrobial agents as control adjuvants have been proposed to reduce the bacterial colonization of tooth surfaces in the post-surgical period [13]. *P. gingivalis*, *T. forsythia*, *T. denticola*, and *A. actinomycetemcomitans* are essential microorganisms in periodontitis. Other microorganisms, such as *E. nodatum*, are also related to periodontal destruction [14]. Many of these species not only colonize periodontal pockets but are also present in oral mucosa, tongue, and tonsils and are commonly detected in saliva [15,16]. Full-mouth disinfection (FMD) therapy includes CHX adjunct to periodontal instrumentation in one or two appointments. However, no evidence exists to establish the FMD approach to provide additional clinical benefits [17]. Otherwise, CHX in the post-surgical period of no regular hygiene has an evident impact on clinical parameters compared with prophylaxis [18] or placebo [19]. Other studies demonstrate the benefit of using CHX post-surgery to reduce the recolonization of periodontal pathogens, demonstrating the establishment of a less mature flora with a predominance of streptococci [4].

HOCl has shown an antimicrobial effect for oral bacteria in preclinic microbiological studies using high concentrations, such as 180 ppm (332.8 uμM), 250 ppm (474 μM) to 500 ppm (948 μM) [7,8], and lower concentrations as 50 ppm (90 μM). [20] However, the higher concentration of HOCl, such as 220 or 330 ppm, did not significantly decrease the minimum inhibitory volume ratio against the microorganisms [20]. HOCl is associated with cellular alterations and stopping the cell cycle due to its oxidizing effect at low concentrations and the formation of chloramines [21]. At low concentrations below 20 μM, HOCl stimulates increased free radical activity against tissues and activates preforms of collagenase-2 and gelatinase-B proteases through the oxidation of thiol groups. [21]. High concentrations can inactivate proteases and the transport of glucose and amino acids, lipopolysaccharides, endo, and bacterial exotoxins; HOCl oxidizes specific cysteine residues in the active site of gingipains such as Rgp and Kgp (*P. gingivalis* cysteine proteases), reducing their damaging potential on tissues [6].

This study evaluated a non-inferiority hypothesis based on similar behavior between protocols. However, an equivalence study required a considerable sample, so an attempt was made to verify at least one hypothesis of no inferiority using specific statistical tests. When the difference is significant with a *p* < 0.05, it is verified that HOCl is non-inferior to CHX.

Although HOCl has low substantivity compared to CHX [22], HOCl demonstrated a significant effect on gram-negative microorganisms associated with periodontitis, as previously reported [4,7]. The HOCl protocol was not inferior to the CHX protocol to avoid recolonization of *T. denticola* at 21 days. *P. gingivalis* recolonization was also similar in the groups. These results are relevant because CHX is the antiplaque substance considered the gold standard.

Neither CHX nor HOCl affected the recolonization of *A. actinomycetemcomitans*. HOCl was not inferior in reducing and recolonizing *A. actinomycetemcomitans* on days 7 and 21. The recolonization of *A. actinomycetemcomitans* is frequent due to the adhesion capacity to epithelial cells employing a specific adhesive protein that subsequently binds to other species of bacteria through coaggregation phenomena [23,24]. Once this microorganism colonizes the supragingival plaque, it moves to the subgingival biofilm, invades the epithelium of the periodontal pocket, and penetrates the underlying connective tissue [25].

Previous research has shown that HOCl has wide-spectrum antimicrobial properties, low or null systemic toxicity, and possible positive effects on cell proliferation [6,25]. Likewise, in vitro studies have shown a potent antiviral effect against SARS-CoV-2 that may favor its use as an antimicrobial agent [26].

This study showed a similar reduction of the mean of PD and CAL in CHX and HOCl protocols. However, in gain >3 mm of CAL, HOCl tended to show a significant reduction compared to CHX. We could hypothesize that reducing the recolonization of periodontal microorganisms may favor tissue healing. However, HOCl has been shown to have an anti-inflammatory effect in atopic dermatitis, and it is favored when combined with taurine [27,28]. The anti-inflammatory effect of HOCl could also be related to the improvement of healing of periodontal tissues, unlike the CHX, which has been associated with a delay in the proliferation of gingival fibroblasts and the production of collagenous and non-collagenous proteins [29,30].

This study introduced a postsurgical brush after seven days in both groups to avoid the deterioration of the experimental substance. The introduction of soft surgical toothbrushes on days 3 to 14 twice daily, adjunct to CHX, is similar to days 14 to 28. An ultrasoft brush may be desirable even early in the postoperative period [31]. Rinsing with CHX causes extrinsic tooth staining and other adverse effects such as calculus build-up, transient taste disturbance, effects on the oral mucosa, and dry mouth [5]. HOCl at a concentration of 0.05% has a significant sensation of dryness at 7 days, 43.7% in the mucous membranes. However, upon lowering the HOCl concentration, it was reduced on day 21, like CHX.

In this study, 62.5% of the patients reported taste alteration for CHX 0.2% and 12.5% for HOCl, much lower for the HOCl protocol. Other studies reported inferior results to ours, with changes in taste in 23% with CHX at two weeks and 25% at four weeks [32]. Previous studies report 47.1% to 80% dental pigmentation with CHX 0.2% between 7 days to six weeks [12,19]. The CHX protocol reducing the concentration of 0.2% to 0.12% evidenced similar dental pigmentation of 62.5% at 21 days. This pigmentation is associated with forming pigmented metal sulfides and dietary factors as modifiers [33]. Surprisingly, the individuals of the HOCl group reported 43% of teeth whitening at 7 days using HOCl at 0.05%, which slightly decreased with the lowest concentration at 21 days at 10%. This effect could be due to an oxidation reaction of the HOCl, similar to that observed with hydrogen peroxide products [34,35]. Oxidizing substances destroy pigments by removing hydrogen while reducing substances’ activities by removing oxygen [36,37]. Future clinical studies should be directed to evaluate the effect of HOCl on dental enamel and lower doses of HOCl to evaluate the effectiveness and the reduction of adverse effects.

## 4. Materials and Methods

### 4.1. Study Design

A triple-blind, non-inferiority randomized controlled trial with two arms was conducted and registered at ClinicalTrials.gov (accessed on 15 December 2019) under n° NCT05952921. This study follows the Consolidated Standards of Reporting Trials (CONSORT) checklist for reporting this clinical trial (CONSORT extension for non-inferiority trials). According to the Declaration of Helsinki on experimentation involving human subjects, the Ethics Committee of the School of Dentistry of Universidad El Bosque (Act #014-2015) approved the study design.

### 4.2. Participants

Thirty-two patients between 20 and 60 years old diagnosed with chronic periodontitis attending the periodontics Postgraduate Clinics of the School of Dentistry of the Universidad Antonio Nariño in Bucaramanga-Colombia between July and December 2019 participated in this study. All members signed informed consent and were instructed about the objectives and possible risks of the study. Participants had to have a minimum of 20 teeth with at least three sites with probing pocket depth (PD) > 5 mm and clinical attachment level (CAL) > 4 mm, radiographic evidence of bone loss, and good general health and required periodontal surgery. Exclusion criteria included smoking, antibiotic therapy, use of NSAIDs in the last four months, pregnancy or lactation, and systemic diseases.

### 4.3. Sample Size

The sample size was determined using a power and sample size calculator for a non-inferiority trial of continuous outcomes from https://sealedenvelope.com/ (accessed on 14 may 2019), based on a significance level (alpha) of 5% and a power (1-beta) of 80%, assuming a non-inferiority margin of 20% of the observed effect size between HOCL and CHX and considering a hypothetical pre-recolonization of 25% in the CHX group and 50% in the HOCL group. The sample size estimates revealed a minimum sample size of 16 subjects per group.

### 4.4. Randomization

Thirty-two voluntary participants were randomly assigned to receive one of two post-surgical protocols after periodontal surgery: (1) a high-concentration rinse with 0.05% HOCl (7 days), followed by 0.025% HOCl (14 days) [8,22]; (2) a high concentration of 0.2% CHX (7 days), followed by 0.012% CHX (14 days) [5]. The participants had no regular oral hygiene and incorporated a post-surgical brush only after day 14 until the end of the study. In this triple-blind study, opaque and sealed envelopes were used for the assignment of each subject; the investigators did not know what type of rinse was assigned to each patient, and the analysis of the results was performed blindly using a coding system that was not disclosed until the analysis was completed. Randomization was generated by computer using Minitab 18 statistical software. A balanced random permuted block method was assigned to the two treatments. A clinical epidemiologist (DDB) realized the randomization table in five blocks. The mouthwashes were masked and indistinguishable in consistency, packaging, and labeling, but the taste was variable.

### 4.5. Clinical Evaluation

The PI, GI, PD, and CAL and subgingival sampling for microbiological analysis were evaluated in the baseline by a calibrated periodontist. GI and PI were dichotomic (presence or absence of changes in gingiva color to clinical observation or the presence or absence of visible plaque evaluated with a periodontal probe). PD was assessed on days 0 after surgery and 90 days after with a North Carolina periodontal probe (Hu-Friedy, Chicago, IL, USA) at six sites per tooth (mesiobuccal, buccal, distobuccal, distolingual, lingual, and mesiolingual), except for the third molar; these sites were also used to assess CAL and BoP (present or absent). On day 7, the suture was removed, and the PI, GI, saliva sample, and subgingival plaque were realized for microbiological analysis. These analyses were repeated at 21 and 90 days.

### 4.6. Microbiological Evaluation

Bacterial samples from the six sites with the greatest PD were collected with sterile paper points size 40 (Maillefer, Dentsply^®^) (Maillefer Instruments Holding SA; Ballaigues; Suiza. Dentsply Sirona; Pennsylvania, PA, USA) for 60 s and introduced into a sterile 1.5 mL tube labeled with the patient’s name. The samples were refrigerated at −20 °C until processing. Tris-EDTA (TE) buffer pH 7.4 buffer was added to the tubes containing the subgingival plaque tips and mixed by vortexing for 20 min. The supernatant was removed and transferred to a 1.5 mL tube for centrifugation at 14.000 rpm for 10 min at 4 °C. The supernatant was discarded, and the pellet was resuspended in 300 µL of sterile deionized distilled water molecular grade. Once homogenized in the vortex, it was frozen at −20 °C overnight. For DNA extraction and subsequent polymerase chain reaction (PCR), a protocol established in the Oral Microbiology Laboratory of the UIBO Institute was used, which consisted of DNA extraction by heat shock. Real-time PCR with absolute quantification allowed confirmation of the number of colony-forming units (CFU). All samples were amplified in a BioRad CFX 96 thermal cycler. The absolute quantification was carried out with the help of calibration curves made for each bacterium with DNA from reference strains with known amounts of bacteria in CFU; the data were transferred to Log10 for statistical analysis.

To identify *P. gingivalis*, primers and the probe reported by Boutaga et al. in 2003 were used [38], previously standardized in our laboratory [39]. To identify *A. actinomycetemcomitans*, the protocol reported by Boutaga et al. in 2005 [40] was used and previously standardized in our laboratory [41].

*T. forsythia* was identified according to the protocol reported by Morillo et al. in 2004 [42]. and *T. denticola* according to the protocol of Yoshida et al. in 2004 [43]. The identification of *E. nodatum*, the protocol previously standardized by our group, was used [44].

### 4.7. Adverse Effects

A survey was carried out for each of the participants at 7 and 21 days of the study to identify clinical adverse effects such as burning sensation, burning or pain in the oral mucosa, sensation of dryness or dryness, and changes in the perception of taste or the color in the teeth. Microbiological adverse effects were performed through saliva samples taken at 0, 7, 21, and 90 days to verify the absence or presence of opportunistic flora associated with mouthwashes.

### 4.8. Statistical Analysis

A descriptive analysis was carried out to compare the groups according to the sociodemographic, clinical, and periodontal status variables. Obtained data were reported as the mean and standard deviation or expressed in median and interquartile range according to the type of distribution based on the Shapiro–Wilk test. PD, CAL, and BoP were compared at baseline and 90 days with paired t-student. Group comparison was performed with a t-student with a significance level of 5% (*p* < 0.05). Repeated measures ANOVA adjusted for treatment–time and time–treatment interaction was used to assess plaque and gingival index at 0, 7, 21, and 90 days. A mixed linear model of repeated measures adjusted for treatment, time, and time–treatment interaction was used. The different bacterial species in times and frequencies of adverse events between groups were determined using a Chi-square/Fisher’s exact test with a significance level of 5% (*p* < 0.05). For the non-inferiority analysis, it was predetermined that HOCL would be considered non-inferior to HOCL rinse administration if the upper boundary of the two-sided 95% confidence interval for the difference between the groups was less than the margin, Δ = −20% [45]. Estimations were conducted using the Hodges–Lehmann hypothesis estimation for non-inferiority with Hodges–Lehmann confidence limits or the hypothesis test for difference in proportions for non-inferiority [46], as appropriate. All analyses were performed using the statistical software programs STATA 14 ((StataCorp LLC; Texas, TX, USA) and Stat Graphics v.18^®^ (Statgraphics Technologies, Inc; Virginia, VA, USA).

## 5. Conclusions

HOCl protocol is not inferior to CHX as a post-surgical antiplaque substance. HOCl reduces the recolonization of periodontal pathogens, showing low adverse effects. Future studies should compare the high and low concentrations of HOCl to establish their differences. HOCl emerges as an alternative for inhibiting dental plaque formation without adverse events or toxicity. Research has shown that due to its antimicrobial (non-antibiotic) properties, low systemic toxicity, and possible effects on cell proliferation, this substance could be used as an antiplaque agent in post-surgical periods.

## 6. Strength and Limitations

This study is the first clinical trial comparing the effectiveness of an antiplaque product of HOCl with CHX as the gold standard. Although the sample size is limited, no inferiority can be demonstrated statistically. The concentration used was based on preclinical and clinical previous studies. HOCl concentrations below 500 ppm throughout the postoperative period should be evaluated to reduce adverse effects further. The discoloration of teeth due to HOCl could not be confirmed clinically.

## Figures and Tables

**Figure 1 antibiotics-12-01311-f001:**
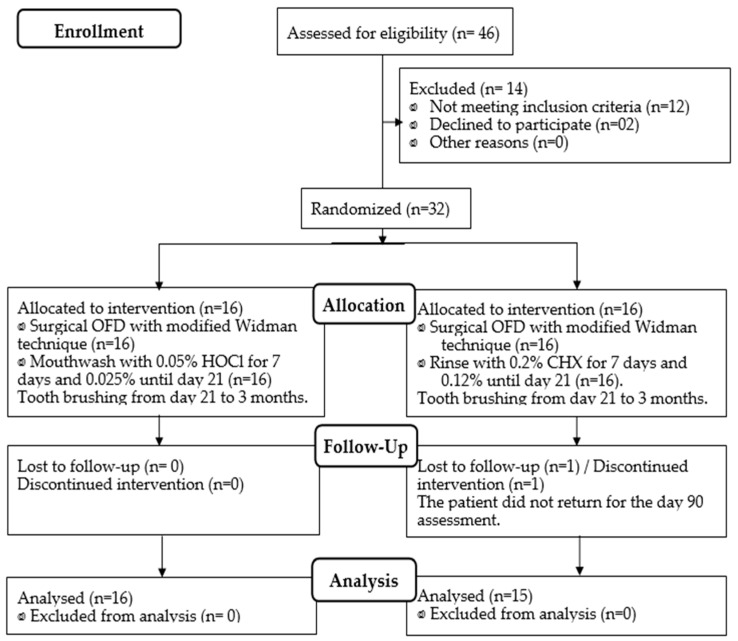
Flowchart of patient allocation (CONSORT 2010).

**Figure 2 antibiotics-12-01311-f002:**
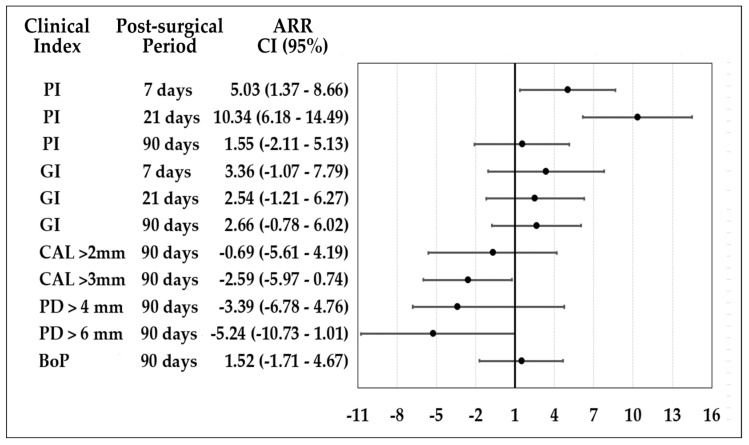
Attributable risk reduction of HOCl and CHX postsurgical protocols.

**Table 1 antibiotics-12-01311-t001:** Baseline sociodemographic and clinical characteristics of the population.

Variables	CHX Protocol	HOCl Protocol	*p*-Value
Gender (F %)			0.273
Female	12 (60.0%)	8 (40.00%)	
Male	8 (66.67%)	4 (33.33%)	
Age (Mean ± SD)	40.4 ± 10.8	40.6 ± 9.4	0.945
Teeth (Mean ± SD)	25.3 ± 2.2	25.1 ± 2.8	0.782
Full Mouth Indexes			
Plaque Index (Mean ± SD)	57 ± 10	57 ± 14	0.956
Gingival Index Mean ± SD)	67 ± 14	62 ± 15	0.777
Bleeding on probing (Mean ± SD)	58 ± 14	56 ± 13	0.607
Pocket Depth (Mean ± SD)	2.51 ± 0.30	2.53 ± 0.30	0.881
Clinical attachment level (Mean ± SD)	2.53 ± 0.92	2.43 ± 0.66	0.725
Experimental Quadrant			
Plaque Index (Mean ± SD)	78 ± 15	70 ± 20	0.205
Gingival Index Mean ± SD)	79 ± 13	71 ± 17	0.169
Bleeding on probing (Mean ± SD)	58 ± 3.5	56 ± 3.2	0.621
Pocket Depth (Mean ± SD)	3.69 ± 0.78	3.67 ± 0.72	0.949
Clinical attachment level (Mean ± SD)	3.49 ± 1.17	3.52 ± 0.76	0.939

Analysis was performed using chi-square and *t*-test.

**Table 2 antibiotics-12-01311-t002:** Comparison of the plaque and gingival indexes between CHX and HOCl over time.

	t_0_	t_1_	t_2_	t_3_
Plaque Index (%)				
CHX (Mean ± SD)	78 ± 15 ^a,b,c^	10 ± 7.9 ^a^	12 ± 7 ^b,†^	10.8 ± 6.6 ^c^
HOCl (Mean ± SD	70 ± 20 ^a,b,c^	15 ± 10 ^a^	22 ± 15 ^b,†^	12.4 ± 9 ^c^
Difference between groups with T0 (IC_95%_)		5.03	10.34 ^†^	1.55
		(1.37–8.66)	(6.18–14.49)	(−2.11–5.13)
Gingival Index (%)				
CHX (Mean ± SD)	79 ± 13 ^a,b,c^	18 ± 8 ^a^	11 ± 0.11 ^b^	8.5 ± 7.4 ^c^
HOCl (Mean ± SD)	71 ± 17 ^a,b,c^	21 ± 13 ^a^	14 ± 12 ^b^	11.5 ± 8.1 ^c^
Difference between groups with T0% (IC_95%_)		3.36	2.54	2.66
		(−1.07–7.79)	(−1.21–6.27)	(−0.78–6.02%)

Repeated measures ANOVA t_0_ = Pre-treatment; t_1_ = Day 7; t_2_ = Day 21; t_3_ = Day 90; ^a^ = differences t_0_ vs. t_1_ *p* < 0.001; ^b^ = differences t_0_ vs. t_2_ *p* < 0.001; ^c^ = differences t_0_ vs. t_3_ *p* < 0.001. Differences between groups: the Hodges–Lehmann Test hypothesis test for non-inferiority. ^†^ *p* < 0.05: non-inferiority.

**Table 3 antibiotics-12-01311-t003:** Comparison between groups for pocket depth, clinical attachment level, and bleeding between baseline and 90 days.

Clinical Index Protocol Groups	t_0_	t_3_	*p*-Value
Probing Pocket Depth			
CHX (Mean ± SD)	3.69 ± 0.78	2.08 ± 0.20	<0.0001
HOCl (Mean ± SD)	3.77 ± 0.63	2.21 ± 0.23	<0.0001
Clinical attachment level			
CHX (Mean ± SD)	3.49 ± 1.17	2.46 ± 0.92	<0.0001
HOCl (Mean ± SD)	3.65 ± 0.57	2.48 ± 0.52	<0.0001
Bleeding on Probing			
CHX (Mean ± SD)	77.1 ± 16.5	7.8 ± 6.7.	<0.0001
HOCl (Mean ± SD)	71.4 ± 17.7	9.9 ± 7.3	<0.0001

Student’s *t*-test, Significant differences *p* < 0.0001. t_0_ = Pre-treatment, t_3_ = Day 90. Differences among the groups *p* > 0.05.

**Table 4 antibiotics-12-01311-t004:** Comparison of concentration of the microorganisms evaluated.

	CHX	HOCl	*p*-Value
	Median IQR	Median IQR	
*P. gingivalis*			
*t*_0_	6.27 (3.93–7.02)	6.84 (3.84–7.37)	0.94
*t*_1_	2.33 (0–3.82)	2.97 (0–3.98)	0.93
*t*_2_	3.27 (0–3.60)	1.28 (0–4.34)	<0.001
*t*_3_	3.16 (0–3.88)	3 (0–4.36)	0.24
*A. actinomycetemcomitans*			
*t*_0_	1.17 (0–3.31)	2.52 (0–2.88)	1.0
*t*_1_	0.00 (0–1.41)	0.00 (0–2.78)	0.31
*t*_2_	2.24 (0–2.94)	2.29 (0–2.73)	0.42
*t*_3_	1.08 (0–2.84)	2.36 (0–2.97)	1.0
*T. forsythia*			
*t*_0_	2.36 (0.21–3.17)	3.48 (1.91–4.11)	0.99
*t*_1_	0 (0–0)	0 (0–0)	N.C
*t*_2_	0 (0–0)	0 (0–0.13)	N.C
*t*_3_	0 (0–0.81)	0.2 (0–2.25)	0.73
*T. denticola*			
*t*_0_	2.36 (0.21–3.17)	3.48 (1.91–4.11)	0.99
*t*_1_	0 (0–0)	0 (0–0)	N.C
*t*_2_	0 (0–0)	0 (0–0.135)	N.C
*t*_3_	0 (0–0.81)	0.2 (0–2.25)	0.73
*E. nodatum*			
*t*_0_	2.69 (0–3.97)	3.4 (0–4.34)	0.94
*t*_1_	0 (0–0)	0 (0–0)	N.C
*t*_2_	0 (0–0)	0 (0–0)	N.C
*t*_3_	0 (0–2.05)	0 (0–0.84)	0.38

Analysis was performed using the Hodges–Lehmann Test hypothesis test for non-inferiority. *p* < 0.05: non-inferiority between groups has been demonstrated. N.C = Not calculable. IQR = Interquartile range (percentile 25–75). t_0_ = Pre-treatment; t_1_ = Day 7; t_2_ = Day 21; t_3_ = Day 90.

**Table 5 antibiotics-12-01311-t005:** Levels of detection, reduction, and colonization of microorganisms in the groups treated with CHX and HOCl over time.

		No Detection			With Detection			Reduction			Recolonization		
	*n*	CHX	HOCl	*p*-Value	CHX	HOCl	*p*-Value	CHX	HOCl	*p*-Value	CHX	HOCl	*p*-Value
*P. ginvgivalis*													
t_0_	16/16	2 (12.5)	3 (18.7)	0.389	14 (87.5)	13 (81.3)	0.114	-	-	-	-	-	-
t_0_–t_1_	16/16	2 (12.5)	1 (6)	0.067	14 (87.15)	15 (94.0)	0.363	8 (50)	7 (43.8)	0.189	6 (37.5)	8 (50)	0.555
t_0_–t_2_	16/16	2 (12.5)	2 (12.5)	0.207	14 (87.5)	14 (87.5)	0.207	9 (56.2)	7 (43.8)	0.112	5 (31.2)	7 (43.8)	0.556
t_0_–t_3_	16/15	2 (12.5)	1 (6)	0.079	14 (87.5)	14 (93.3)	0.353	7 (43.7)	7 (46.6)	0.352	7 (43.7)	7 (46.6)	0.292
*A. actinomycetemcomitans*													
t_0_	16/16	8 (50.0)	5 (31.2)	0.056	8 (50.0)	11 (68.7)	0.056	-	-		-	-	
t_0_–t_1_	16/16	8 (50.0)	6 (37.5)	0.110	8 (50.0)	10 (62.5)	0.110	4 (25)	1 (6)	0.016	4 (25)	9 (56.2)	0.010
t_0_–t_2_	16/16	7 (43.7)	5 (31.2)	0.105	9 (56.2)	11 (68.7)	0.105	5 (31.2)	2 (12.5)	0.029	4 (25)	9 (56.9)	0.010
t_0_–t_3_	16/15	7 (43.7)	5 (31.2)	0.105	9 (56.2)	10 (66.5)	0.133	6 (37.5)	4 (26.6)	0.118	3 (18.7)	6 (40)	0.705
*T. forsythia*													
t_0_	16/16	4 (25.0)	3 (18.7)	0.144	12 (75.0)	13 (81.2)	0.402	-	-	-	-	-	-
t_0_–t_1_	16/16	4 (25.0)	3 (18.7)	0.144	12 (75.0)	13 (81.2)	0.402	1 (6.2)	0 (0)	0.276	11 (68.7)	13 (81.2)	0.563
t_0_–t_2_	16/16	3 (18.7)	3 (18.7)	0.244	13 (86.6)	13 (81.2)	0.244	1 (6.2)	3 (18.7)	0.583	12 (75)	10 (62.5)	0.095
t_0_–t_3_	16/15	3 (18.7)	3 (20)	0.278	13 (86.6)	12 (80.0)	0.278	3 (18.7)	5 (33.3)	0.563	10 (62.5)	7 (46.6)	0.084
*T. denticola*													
t_0_	16/16	4 (25)	3 (18.7)	0.144	12 (75)	13 (81.2)	0.402	-	-	-	-	-	-
t_0_–t_1_	16/16	4 (25)	3 (18.7)	0.144	12 (75)	13 (81.2)	0.402	1 (6.2)	0 (0)	0.276	11 (68.7)	13 (81.2)	0.563
t_0_–t_2_	16/16	3 (18.7)	3 (18.7)	0.244	13 (86.6)	13 (86.6)	0.244	1 (6.2)	4 (26.6)	0.016	13 (81.2)	9 (56.9)	0.020
t_0_–t_3_	16/15	3 (18.7)	3 (20.0)	0.278	13 (86.6)	12 (80.0)	0.278	3 (18.7)	5 (33.3)	0.068	10 (62.5)	7 (46.6)	0.084
*E. nodatum*													
t_0_	16/16	7 (43.7)	5 (31.2)	0.104	9 (56.2)	11 (68.7)	0.556	-	-	-	-	-	-
t_0_–t_1_	16/16	4 (25.0)	3 (18.7	0.144	12 (75.0)	13 (86.6)	0.402	1 (6.2)	0 (0)	0.276	11 (68.7)	13 (81.2)	0.563
t_0_–t_2_	16/16	7 (43.7)	5 (31.2)	0.104	9 (56.2)	11 (68.7)	0.556	2 (13)	3 (18.7)	0.389	7 (43.7)	8 (50)	0.419
t_0_–t_3_	16/15	4 (25)	4 (26.6)	0.306	12 (75.0)	12 (81.2)	0.375	0 (0)	2 (13)	0.646	12 (80)	10 (66.6)	0.142

Analysis was performed by testing the difference in proportions hypothesis for non-inferiority. *p* < 0.05: non-inferiority between groups has been demonstrated. t_0_ = Pre-treatment; t_1_ = Day 7; t_2_ = Day 21; t_3_ = Day 90.

**Table 6 antibiotics-12-01311-t006:** Adverse events on days 7 and 21 according to the antimicrobial protocol.

	7 Days					21 Days				
	CHX		HOCl		*p*-Value	CHX		HOCl		*p*-Value
	*n*	(%)	*n*	(%)		*n*	(%)	*n*	(%)	
Taste Sensation										
Pleasant	5	31.3	1	6.3	0.097	6	37.5	1	6.3	0.048
Unpleasant	10	62.5	15	93.8		9	56.3	15	93.8	
Disgusting	1	6.3	0	0.0		1	6.3	0	0.0	
Mouth irritation										
Absence	6	37.5	6	37.5	1.000	11	68.8	14	87.5	0.394
Presence	10	62.5	10	62.5		5	31.3	2	12.5	
Pain										
Absence	16	100.0	16	100.0	1.000	16	100.0	16	100.0	1.000
Presence	0	0.0	0	0.0		0	0.0	0	0.0	
Burning										
Absence	12	75.0	15	93.8	0.144	15	93.8	15	93.8	1.000
Presence	4	25.0	1	6.3		1	6.3	1	6.3	
Numbness										
Absence	9	56.3	8	50.0	0.723	14	87.5	13	81.3	1.000
Presence	7	43.8	8	50.0		2	12.5	3	18.8	
Burning sensation										
Absence	15	93.8	16	100.0	0.310	16	100.0	16	100.0	1.000
Presence	1	6.3	0	0.0		0	0.0	0	0.0	
Roughness										
Absence	13	81.3	14	87.5	0.626	16	100.0	15	93.8	0.310
Presence	3	18.8	2	12.5		0	0.0	1	6.3	
Lip Dryness										
Absence	13	81.3	9	56.3	0.127	11	68.8	10	62.5	0.710
Presence	3	18.8	7	43.8		5	31.3	6	37.5	
Change in taste sensation										
Absence	9	56.3	14	87.5	0.049	6	37.5	14	87.5	0.009
Presence	7	43.8	2	12.5		10	62.5	2	12.5	
Gastric Alteration										
Absence	16	100.0	16	100.0	1.000	16	100.0	16	100.0	1.000
Presence	0	0.0	0	0.0		0	0.0	0	0.0	
Color change in teeth										
Absence	13	81.3	9	56.3	0.127	5	31.3	11	68.8	0.034
Presence	3	18.8	7	43.8		11	68.8	5	31.3	
Color change trend										
White	16	100.0	15	93.8	0.310	0	0.0	5	100.0	0.000
Black	0	0.0	1	6.3		11	100.0	0	0.0	
Location of the AE										
No AE	0	0.0	0	0.0	0.081	10	62.5	9	56.3	0.083
Lips	7	43.8	12	75.0		0	0.0	0	0.0	
Gum	2	12.5	0	0.0		0	0.0	4	25.0	
Palate	2	12.5	1	6.3		0	0.0	1	6.3	
Language	0	0.0	2	12.5		3	18.8	0	0.0	
Whole mouth	5	31.3	1	6.3		3	18.8	2	12.5	

## Data Availability

Not applicable.

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
