# Peer review of "Hypochlorous Acid as a Potential Postsurgical Antimicrobial Agent in Periodontitis: A Randomized, Controlled, Non-Inferiority Trial"

_antibiotics, 2023, doi:10.3390/antibiotics12081311_

Round 1

Reviewer 1 Report

The article by Julio et al. report a study that compares the effects of hypochlorous acid (HOCl) and chlorhexidine (CHX) rinses after periodontal surgery in patients with chronic periodontitis. The study finds that HOCl is not inferior to CHX in reducing plaque, gingivitis, pocket depth, and clinical attachment level, and in preventing the recolonization of periodontal pathogens1. The article also discusses the advantages of HOCl as a potential antimicrobial agent with low toxicity and possible anti-inflammatory and proliferative cell effects2. The article concludes that HOCl could be used as an alternative or adjunct to CHX in periodontal therapy. The manuscript is well-written and well-organized. However, there are several areas in the manuscript that require improvement. 

The authors might have to refine their figures to ensure a consistent font format, font size, and color to enhance their appearance and readability.

The authors employed two different concentrations of the agents: 0.05% and 0.025% for HOCl, and 0.2% and 0.012% for CHX. The ratio between the high and low concentrations of HOCl is 2, while for CHX, it is approximately 16.7. It is essential for the authors to provide a rationale for selecting these specific concentrations.

 Minor editing of English language required

Author Response

Response: HOCl and CHX have different concentrations because they are different molecules.  However, CHX exists in two concentrations for clinic use CHX 0.2%, considered a higher dose, and 0.12%, half the concentration, and a lower dose. HOCl is similar; the higher dose is 0.05% (500 ppm), and the half concentration is 0.012% (250 ppm).  Both substances and their concentration has been established in preclinic and clinic study.  In the text, we included references to these studies for HOCl and CHX. The references to Clinical and preclinical studies were included (See page 2, line 56; page 11, line 334).

The figures were adjusted according to the indications (font format, font size, and color) to improve their visualization.

Minor edition of the English language:

Response: This was done.

Reviewer 2 Report

In the manuscript by Plata et al, the author compared the Hypochlorous acid (HOCl) with Gold standard Chlorhexidine as a postsurgical antimicrobial agent in Periodontitis in a no-inferiority randomized controlled trial.  Following flap surgery and authors used two antimicrobial protocols and compared the reduction of plaque index (PI), gingival index (GI), pocket depth (PD), gain of clinical attachment level (CAL), and bacterial recolonization in subgingival biofilm at 7, 21, and 90 days. The study is well designed with all the parameters required for the comparative evaluation of two antimicrobial treatments. The manuscript is well drafted, and the authors did a good job using statistical tools to compare the equivalent or no inferiority effect of HOCl and CHX. I appreciate the authors for writing a good discussion section as well.  Authors are advised to do the suggested modifications and justify the queries.

Comments to the Author:

1. Why the authors used a specific concentration of HOCl (0.05% followed by 0.025%)? Any justification?

2. Authors should mention some previous research and review on HOCl in periodontitis in the Introduction (3RD Paragraph Page No 2). There were some excellent articles on HOCl and periodontitis.

Aherne O, Ortiz R, Fazli MM, Davies JR. Effects of stabilized hypochlorous acid on oral biofilm bacteria. BMC Oral Health. 2022 Dec;22(1):1-2.

Sam CH, Lu HK. The role of hypochlorous acid as one of the reactive oxygen species in periodontal disease. Journal of Dental Sciences. 2009 Jun 1;4(2):45-54.

Castillo DM, Castillo Y, Delgadillo NA, Neuta Y, Jola J, Calderón JL, Lafaurie GI. Viability and effects on bacterial proteins by oral rinses with hypochlorous acid as the active ingredient. Brazilian dental journal. 2015;26:519-24.

2.  Please use italics for all microorganisms’ names throughout the manuscript.

3.  Page 8 Line 204 “CHX is more effective in 10% in reduces PI than HOCl at seven days”. Please reframe this sentence. Replace the word “reduces” with “reducing”

Author Response

Comment 1: Why the authors used a specific concentration………………………………

Response: HOCl has been used at 0.05% (500 ppm) in clinical medicine and studied in clinical and preclinic studies, including microbiologic and cytotoxic studies. HOCl 0.05% and 0.025% (250 ppm) were studied by our group in previous preclinic and clinic studies.  Like CHX, its use in lower doses has been evaluated to reduce adverse effects. In the text, we included references to previous studies for HOCl in both concentrations. The references to Clinical and preclinical studies of HOCl were included (See page 2, line 56; page 11, line 334). (See page 11, lines 317).

Comment 2: Authors should mention some previous research and review…….

Response: These references, Aherne et al., 2022 and Castillo et al., 2015 are included in the introduction, and these are original studies on periodontal biofilm. However, the text slightly expanded the results (See page 2, lines 52-60).  Sam et al., 2009 are included (see reference 6).

Comment 3: Please use Italic …….

Response: This was done

​Comment 4: Page 8, Line 204………..

Response: This was done

Reviewer 3 Report

In this study, authors compared the effects of two antimicrobial protocols, HOCl 0.05% followed by HOCl 0.025%, and CHX 0.2%/CHX 0.12%, on reducing plaque index, gingival index, pocket depth, gain of clinical attachment level, and bacterial recolonization in subgingival biofilm after Open Flap Debridement (OFD) in patients with periodontitis. The results showed that HOCl was not inferior to CHX in reducing plaque index and both protocols had a comparable reduction of certain periodontopathic bacteria in the postoperative period.

The manuscript fit the scope of the journal and special issue. Before publication some minor improvements are necessary:

1.      Highlight the limitation (eg. number of patients) and novelty of the study. HOCL is a well-known agent. Just see Ref [11].

2.      The disparity in gender ratios between the two groups is conspicuous.

3.      Check line 31 and 35 in Abstract. It is a repetation.

4.      HOCL is not a good keyword. I suggest changing it

5.      More information regarding HOCL is necessary.

6.      In Figure 1 2 instead of 02.

7.      Page 4 Line 90 t0, t3 etc. I suggest t0, or it should be uniform. Somewhere you write T2 etc.

Author Response

Comment 1: Highlight the limitations……….

Response:  The limitations and novelty were included in the text in “Strength and Limitations” (see page. 13, lines 416-422).

Comment 2: Differences in gender are…..

Response: These data were changed by an error when transcribing the numbers. Thanks for your correction.

Comment 3: Ckeck line 31 and 35……

Response: This was corrected.

Comment 4: HOCl is not a good keyword.

Response: This word was removed,

Comment 5: More information regarding HOCl is necessary…..

Response: In the introduction, HOCl concepts were expanded. (See page 2, lines 52-60)

Comment 6: In addition, the numbering of the figures was validated (Figure 1, Figure 2).

Comment 7: Page 4. Line 90 I suggest …….

Response: This was done according to your suggestion.

Reviewer 4 Report

The authors wanted to compare the effectiveness of Hypochlorous acid with respect to chlorhexidine on some parameters in patients with periodontitis who underwent open flap debridement in their study. The work is generally well designed and well written.

I don't understand why the method section in the article was put after the discussion. I will not say anything if it is the request of the journal. But when I read, I habitually prefer to read the method section before the results.

In the Discussion section, it was stated that Hypochlorous acid was used in some parameters in previous studies. In this section, I think that it will add value to the study to talk about the difference between this study and other studies in the literature, as well as the features that distinguish this study from the studies in the literature, or the advantages, if any.

When I look at the literature, I see that this issue has been studied in different ways. I believe that the authors should emphasize the features that distinguish their work from other studies in the literature in the conclusion section.

Kind regards

Author Response

Comment 1: I don´t understand why the methods section ……

Response: These are the indications of the journal, and we cannot change the methods' place.

Comment 2:In the discussion section, it was stated that hypochlorous ……

Response: Results were discussed with other preclinic studies (see page 9, lines 238-.250)

Comment 3: When I look al the literature, I see this issue ………..

Response: This was included in strengths and limitations because did not exist another randomized clinical trial which evaluated HOCL with CHX.